# Tribological Performance of Micro-Groove Tools of Improving Tool Wear Resistance in Turning AISI 304 Process

**DOI:** 10.3390/ma13051236

**Published:** 2020-03-09

**Authors:** Jinxing Wu, Gang Zhan, Lin He, Zhongfei Zou, Tao Zhou, Feilong Du

**Affiliations:** 1School of Mechanical Engineering, Guizhou University, Guiyang 550025, China; gs.pftian17@gzu.edu.cn (J.W.); gz_zhoutao@163.com (T.Z.); amt.fldu@gzu.edu.cn (F.D.); 2School of Mechanical and Electrical Engineering, GuiZhou MinZu University, GuiYang 550025, China; 3Department of automotive engineering, Guizhou Vocational Technology College of Electronics & Information, Kaili 556000, China; zhangangbmw@163.com; 4School of Mining & Civil Engineering, Liupanshui Normal Colleague; Liupanshui 553004, China; 5School of Mechanical Engineering, GuiZhou Institute of Technology, GuiYang 550003, China; 15285127055@163.com

**Keywords:** micro-groove tool, friction, cutting force, wear

## Abstract

AISI 304 has good physical and chemical properties and thus is widely used. However, due to the low thermal diffusivity, the cutting temperature of AISI 304 is high accelerating the wear of the tool. Therefore, tool wear is a major problem in machining hard cutting materials. In this study, we developed a new type of micro-groove tool whose rake surface was distributed with micro-groove by powder metallurgy based on the finite element temperature field morphology. We compared the wear of the proposed micro-groove tool with an untreated one by using a scanning electron microscope (SEM) and an X-ray energy spectrum. The abrasive, adhesive, and oxidation wear of the rake and the flank face of the micro-groove tool were lower than that of the untreated one. Due to the micro-groove on the rake face of the tool, the contact length between the tool and chip was reduced, leaving more extension space. Furthermore, chip extrusion deformation was avoided, and the energy caused by chip deformation was reduced. After 70 min of cutting, the counterpart reached the specified wear amount while the main cutting force, the feed resistance, and the cutting depth resistance of the proposed micro-groove tool were reduced by 16.1%, 33.9%, and 40.1%, respectively. With regard to steady state, the cutting temperature was reduced by 17.2% and the wear width of the flank face was reduced by 36.7%.

## 1. Introduction

AISI 304 has good comprehensive performance in high-temperature, corrosive environments and it is widely used in important fields such as shipping, aviation, medicine, and nuclear power. However, because of its good toughness, high plasticity, and serious work hardening during cutting, the tool is prone to wear and its cutting life is short. To solve such problems, methods involving cutting fluids, lubrication systems, minimum quantity lubrication (MQL), and tool surface coatings are generally used to improve the tribological properties of the tool–chip contact area.

The cooling lubricant can reduce the friction at the tool–chip interface and the heat generated during the cutting process. It can also reduce the normal and shear stress of the tool–chip interface as well as the stress distribution along the interface [1,2]. Several scholars have studied the relationships among dry cutting, MQL, and minimum quantity cooling lubrication (MQCL). Compared to dry cutting and conventional machining, MQL and MQCL reduce the cutting zone temperature and tool wear, consequently prolonging tool life [3,4,5]. Maruda et al. applied a cemented carbide tool to cut AISI 1045 and found that MQCL with extreme pressure anti-wear additive can form a thin friction film layer on the tool surface during the cutting process, enhancing the wear resistance of the tool [6]. Hegab et al. used a nano-additive to improve the cold cutting efficiency of titanium alloy on the basis of micro-lubrication [7]. Compared with micro-lubrication, nanofluids improve the lubrication property of the rake and flank face of the tool, enabling it to have better cutting performance and better energy consumption performance. Jamil et al. found that Al_2_O_3_ Multiwalled carbon nanotube-based MQL noticeably reduced the surface roughness and main cutting force compared with cryogenic CO_2_-based cooling [8]. Hegab et al. used multiwalled carbon nanotubes and aluminum oxide as nano-cutting fluid additives to cut Inconel 718 and found that these two kinds of nano-fluid improve the tool wear [9]. When cutting AISI 1045 steel, Adel et al. found that using the cutting method of MQL-assisted nanofluid can obtain better machining surface quality and lower energy consumption [10].

In recent years, dry cutting has gradually become a new trend [11]. However, this method has greater requirements for high temperature resistance and wear resistance of the tool. Fortunately, the tool coating is just the barrier of the cutting heat and chemical corrosion, thus reducing the impact of heat on the tool. The tool coating can also reduce the friction of the tool–chip interface, optimizing the dry cutting environment [12,13].

Many researchers have shown that physical vapor deposition (PVD) coatings like CrN, AlCrN, TiN, and AlTiN can improve the cutting performance of the tool as well as prolong its life [14,15,16,17,18,19]. For example, Beake et al. found that AlCrN and AlTiN coating tools performed well under extreme conditions because of their excellent oxidation resistance, high-temperature hardness, and high-temperature wear resistance [20]. Some scholars have also studied the tribological properties of single and double coatings. For single-layer coatings, TiN has a lower friction coefficient and TiAlN has a lower wear resistance [21,22]. The TiN/TiAlN multi-layer coating has a lower friction coefficient, lower wear rate, and better corrosion resistance than the single-layer coating [23].

The large use of cutting fluids increases the cost as well as the pressure of the environment. However, it is difficult to develop new coating techniques as the current ones have become very well-developed. Therefore, tool texture alteration has proven to be an effective method for improving the cutting performance of the tool. Kawasegi et al. experimentally studied the cutting properties of A5052 aluminum alloy by using femtosecond lasers to generate micro-scale and nanoscale textures on the rake face of carbide tools [24]. It was found that the cutting force and friction coefficient of the textured tool decreased along the tool–chip interface during the cutting process. This decrease in the cutting force and friction coefficient depends on the direction of the texture. Koshy et al. used electric sparks to produce isodirectional grooves on the rake face of tool [25]. With such a structure, the cutting force and feed resistance were significantly reduced. In addition, large, deep grooves could be used to more effectively improve tribological properties [26].

Kümmel et al. used lasers to form different textures on the surface of the tool by which the tool adhesion can be changed by specific textures [27]. Deng et al. tested three surface texture tools with different geometric characteristics (elliptical, parallel, and vertical grooves) in which the elliptical one had the lowest cutting force, cutting temperature, and friction coefficient of the tool–chip interface [28]. Based on the literature, it is clear that tool textures can improve cutting performance and reduce tool wear. However, the texture shape and size are mostly based on bionics or multiple experiments, lacking theoretical basis. In this study, we cooperated with an enterprise (Zigong Cemented Carbide Tool Corp., Ltd., Zigong, China) aiming to develop a tool that can be used to process medical ware. We proposed a micro-groove design method by simulating the movement and temperature distribution of the rake face during the cutting process. Such a design was based on a focus on reduction of the friction coefficient and cutting energy. In this way, two cutting tools were designed for cutting 40CrMnMo and Inconel 718. We compared the cutting force, cutting temperature, tool–chip frictional coefficient, sliding energy, and wear of the rake and flank face of the proposed tool with the untreated one. We believe that the micro-groove tool can improve the distribution of cutting energy and reduce tool wear during the cutting process [29,30]

The second section describes the design and manufacture of cutting tools. The third section describes the experimental methods and materials. The fourth section is the experimental results and discussion, including the calculation of knife chip friction coefficient and friction energy and the analysis of SEM and X-ray energy spectrometer on the rake and flank face of the micro-groove tool. Last, we present the conclusion.

## 2. Materials and Methods

### 2.1. Tool Design and Manufacturing

In this study, commercial cemented carbide tools (ZiGong Cemented Carbide Tool corp., LTD., Zigong, China) were used for cutting; these were collectively categorized with the label “Tool A” (untreated tools). Tool B was designed by adding a micro-groove near the cutting edge on the rake face of Tool A, which was based on the simulation of the temperature field of Tool A during the cutting process. Tool A and Tool B have the same geometric angles which are listed in Table 1. Both Tools A and B are made of K20 cemented carbide, and the main component is WC. The tool radius is 27.92 μm. 

The design process of Tool B was as follows. First, a cutting simulation was conducted with the use of DEFORM 11.2 software (V11.2, Scientific Forming Technologies Corporation, Columbu, OH, USA), using the cutting parameters recommended by the manufacturer (cutting speed v equals 120 m/min, feed rate f is 0.15 mm per revolution (mm/rev), and cutting depth a_p_ is 1.5 mm). After the simulation, the temperature points and corresponding coordinate data of the high temperature range of the untreated tool were extracted and then processed with MATLAB (R2016b, the Mathworks Inc., Natick, MA, USA). Considering the reliability and machinability of the tool, the temperature field area of the rake face in the range of 540 to 713 °C was selected. After interpolation with MATLAB, the obtained data were imported into NX. Then, the temperature field curve and the mesh surface were fitted by using a nonuniform rational basis spline (NURBS). Finally, the micro-groove was formed by trimming the body function. This established the geometric model of Tool B. Next, Tool B was fabricated with the same powder metallurgy process as that of Tool A. The TiAlN coating was deposited on both tools by PVD with the thickness of 5 μm with 32% Ti, 36% Al, and 32% N. The shape and size of Tool B are shown in Figure 1. The maximum depth of the micro-groove is 0.2 mm. 

### 2.2. Cutting Process

Machining tests were carried out on a C6136HK CNC lathe (the FAVGOL, Chongqing, China). A dynamometer (Kistler-9257-B, Kistler Instrumente AG, Winterthur, Switzerland) was used for testing the cutting force (Figure 2). The workpiece was an AISI 304 cylindrical rod material with a diameter of 60 mm and a length of 150 mm. Material properties and chemical composition of the tool and the workpiece are shown in Table 2 and Table 3.

During the cutting process, we used Tool A and Tool B to cut the same length of the workpiece each time and then recorded the time. 

To intuitively explain the equilibrium force system on the cutting surfaces of Tool A and Tool B, a two-dimensional cutting mechanics model was established, as shown in Figure 3. In this figure, *F_f_* and *F_n_* are the friction force and positive pressure of the tools acting on the chip, respectively, whereas *F_x_* and *F_y_* are the vertical and horizontal components of the workpiece acting on it, respectively. The resultants of these two pairs of forces, *F_rγ_* and *F_rs_*, balance each other.

Because of the existence of micro-groove, the original balance relationship between Tool B and the chips was destroyed. The separation point of the tool and the chip moved along the inner wall of the micro-groove to a certain point, and a new tool–chip balance relationship was established. At this point, the resultant action line rotated in a clockwise direction and the positive pressure direction of the chip and the rake face changed. Following the cutting theory and test comparison, the front angle of Tool B *γ*_0_ increased, the friction angle *β* decreased, and the shear angle *φ* increased (Figure 4B). Assuming that the positive pressure *F_n_* of the cutter to the chip remained unchanged, it could be concluded that *F_x_* and *F_rγ_* would decrease according to Equations (1)–(3).
(1)$$F_{n}=F_{r\gamma }\cos\beta$$
(2)$$F_{x}=F_{r\gamma }\sin(\beta -\gamma_{0})$$
(3)$$F_{y}=\sqrt{F_{r\gamma }^{2}-F_{x}^{2}}$$

To explore the friction status of the tool–chip interface and to compare the changes of the friction factor and the frictional dissipation energy in the cutting process, we built a three-dimensional bevel cutting model (Figure 4A).

According to the theory of metal cutting and the geometric relationship, we could obtain:(4)$$\tan\gamma_{n}=\tan\gamma_{0}\cos\lambda_{s}$$
(5)tanφn=bbccosγn1−bbcsinγn
(6)$$\cos\eta_{c}=t_{c}\cos\lambda_{s}/a_{w}$$
(7)$$\tan\eta_{s}=\frac{\tan\lambda_{s}\cos(\varphi_{n}-\gamma_{n})-\tan\eta_{c}\sin\varphi_{n}}{\cos\gamma_{n}}$$
where *λ_s_*, *φ_n_*, *b*, *b_c_*, *t_c_*, and *a_w_* were, respectively, the inclination angle, the normal shear angle, the cutting thickness, the chip thickness, the chip width, and the uncut width. *γ*_0_ and *γ_n_* were rake and normal angles. *η_c_* and *η_s_* represent the chip angle and shear chip angle. In the cutting process, the three velocity vectors formed closed triangles which can be seen in Figure 4B. According to the three-dimensional model of the velocity triangle, we obtained:(8)$$v_{c}=\frac{\cos\lambda_{s}\sin\varphi_{n}v}{\cos\eta_{c}\cos(\varphi_{n}-\gamma_{n})}$$
(9)$$v_{s}=\frac{\cos\lambda_{s}\sin\varphi_{n}v}{\cos\eta_{s}\cos(\varphi_{n}-\gamma_{n})}$$
where *v_c_*, *v*, and *v_s_* were chip velocity, cutting velocity, and shear velocity, respectively. The equivalent transformation was achieved by angle transformation between the three-dimensional model and the right-angle cutting model. To obtain the three-dimensional coordinate forces (*F_x_’*, *F_y_’*, *F_z_’*), we first made (*F_x_*, *F_y_*, *F_z_*) rotate at angle *γ_n_* around axis *x’*. We then rotated Angle *λ_s_* around axis *z*, as shown in Figure 5.
(10)$$\left(\begin{array}{c} F_{x}^{\prime }\\ F_{y}^{\prime }\\ F_{z}^{\prime } \end{array}\right)=\left(\begin{array}{ccc} 1 & 0 & 0\\ 0 & \cos\gamma_{n} & \sin\gamma_{n}\\ 0 & -\sin\gamma_{n} & \cos\gamma_{n} \end{array}\right)\left(\begin{array}{ccc} \cos\lambda_{s} & -\sin\lambda_{s} & 0\\ \sin\lambda_{s} & \cos\lambda_{s} & 0\\ 0 & 0 & 1 \end{array}\right)\left(\begin{array}{c} F_{x}\\ F_{y}\\ F_{z} \end{array}\right)$$

The friction force and the normal force on the rake face of the tool were:(11)Ff=Fx′2+Fz′2=((Fxcosλs−Fysinλs)2+(−Fxsinλssinγn−Fycosλssinγn+Fzcosγn))12
(12)$$F_{n}=F_{x}\sin\lambda_{s}\cos\gamma_{n}+F_{y}\sin\gamma_{n}+F_{z}\cos\lambda_{s}\cos\gamma_{n}$$

According to Equations (11) and (12), the friction factor of the cutting tool’s rake face was calculated as:(13)$$\mu =\frac{F_{f}}{F_{n}}$$

The frictional energy was obtained from Equations (8) and (11) and the cutting parameters mentioned above:(14)$$E_{f}=\frac{F_{f}v_{c}}{va_{p}f}$$
where *a_p_* and *f* represent cutting depth and feed, respectively.

After the cutting process, the tool was removed and cleaned by an ultrasonic cleaning machine before it was blown dry. Then we used an industrial microscope (BX51-P, Olympus Corporation, Tokyo, Japan) for observation of the cumulative wear amount on the flank face until it reached 150 μm. At the same time, a dynamometer (Kistler-9257-B, Kistler Instrumente AG, Winterthur, Switzerland) was used to test the cutting forces in three cutting directions and the wear mechanism of the two tools was analyzed. Finally, SEM (Zeiss SUPRA 40; Carl Zeiss AG, Jena, Germany) and energy dispersive X-ray spectroscopy (EDS) were used to analyze the wear morphology and element composition of the tool surface.

It was very difficult to obtain the surface temperature of the tool in real time during the cutting process because the position of the tool tip frequently changed. Therefore, we simulated the cutting process with Deform V11.2 software (Scientific Forming Technologies Corporation, Columbus, the USA). In cutting simulation, it is reasonable to raise the heat transfer coefficient to make the tool temperature reach the constant state in a short time, because this is consistent with the experimental results [31]. In the simulation process, the cutting speed *v**c* was 120 m/min, the depth of cut *a**_p_* equaled to 1.5 mm, and the feed Rate *f* was 0.1 mm per revolution (mm/rev). The material model of Tools A and B is John cook model. We set the environmental temperature as 20 °C and the shear friction factor was 0.4. No cutting fluid was added during the cutting process. The convective heat transfer coefficient between the tool and the workpiece was set at 2000 to quickly obtain the dynamic balance and relatively stable tool temperature field of the tool. The tool was regarded as a rigid body and the workpiece as a plastic body, both of which adopted a tetrahedral mesh with a relative mesh size of 50,000 and a dimension ratio of 7. The cutting length was set at 100 mm; simulation results are shown in Figure 6. During the time of 0.006–0.008 s, the temperature of Tool A and Tool B reaches the stable state, and the temperature fluctuation of tools is small. Their average temperatures were 713.2 and 590.5, respectively. The temperature of cutting decreased by 17.2% for Tool B compared to Tool A. 

## 3. Results

### 3.1. Experimental Results

Both Tool A and Tool B were used to cut AISI 304. The flank face reached the specified wear width at 70 min and 110 min for Tool A and Tool B, respectively. Figure 7A shows the wear morphology changes of the rake and flank face of Tool A and Tool B, as observed by the use of industrial microscope BX51-P (Olympus Corporation, Japan). The contrast curves of the wear width of the flank faces are shown in Figure 7B. The tool wear process went through three distinct stages: the initial wear stage, the normal wear stage, and the severe wear stage [32]. As shown in Figure 7A, during the first 20 min, Tool A underwent the initial wear phase. The flank face wore quickly and the corresponding wear reached 110 μm. Then, the tool wear tended to be gentle during the next half hour, up to the 50 min mark. When it approached minute 50, the wear width of the flank face was ~132 μm. As can be seen in Figure 7A, the near-field wear of cutting edges on the rake and flank faces was serious. In the stage of severe wear, the wear on the flank face increased rapidly to 150 μm. In addition, edge breakage occurred at the cutting edge which was far away from the tip for Tool A. For Tool B, in the initial wear phase, the wear amount on the flank face quickly reached 65 μm, which was much less than that of Tool A at the same time. During the normal wear phase, Tool B lasted for nearly 70 min from minute 20 to minute 90 (Figure 7B). The wear amount of the flank face of Tool B was 85 μm and 95 μm when it came to minute 50 and minute 70, respectively. Such a result was also significantly less than that of Tool A. From 90 min to 110 min, Tool B wore quickly and the wear width of the flank face reached 150 μm. Tool B had better wear resistance than Tool A in terms of cutting time and wear width of the flank faces. We carried out this experiment for three times and the statistic results are shown in Table 4, Table 5, Table 6 and Table 7.

### 3.2. Wear Mechanism Analysis 

AISI 304 has good toughness, high plasticity, low thermal conductivity, and severe work hardening. During the cutting process, the stress and temperature of the contact area of the tool–chip was high. Abrasive wear, adhesive wear, and oxidation wear occur easily on the rake and flank faces. Based on the cutting test of tool durability, Tool A and Tool B were cut for 70 min and 110 min, respectively, with the flank face wear reaching a width of 0.15 mm as the limit. SEM and EDS analysis were carried out on the rake and flank faces of both tools, and the results are as follows.

During the cutting process, Kistler-9257B was applied to test the cutting forces of Tool A and Tool B and to calculate the cutting force *F_r_*, $F_{r}=\sqrt{F_{x}^{2}+F_{y}^{2}+F_{z}^{2}}$. The measurement results are shown in Figure 7C. Both the component cutting force and the cutting resultant force of Tool B were reduced, which is consistent with Equations (1)–(3).

It can be seen from Figure 7C that during the cutting process, the mean values of the feed resistance *F_x_*, the main cutting force *F_y_*, the cutting depth resistance *F_z_*, and the cutting force *F_r_* of Tool B were all smaller than those of Tool A. The feed resistance and the cutting depth resistance were reduced by 27.3% and 15.1%, whereas the main cutting force and the cutting force were reduced by 1.3% and 5.7%, respectively.

The cutting force values of Tool A and Tool B tested by dynamometer are shown in Figure 8. Using Equations (1)–(14), the comparison of the friction coefficient and the sliding energy over time were obtained, which are shown in Figure 9A,B. Compared with Tool A, both the friction coefficient and the sliding energy of Tool B were smaller during the whole cutting process, especially after 40 min.

As can be seen from Figure 8 and Figure 9, the main cutting force was almost equal before minute 43. However, the frictional coefficient as well as the sliding energy of Tool A was higher than that of Tool B. In 43 min through 70 min, the main cutting force of Tool A changed rapidly and the flank face of Tool A reached the specified wear width. At the same time, the frictional coefficient as well as the sliding energy of Tool A increased sharply. In contrast, no matter the cutting force, the frictional coefficient or the sliding energy of tool B changed smoothly. From 70 to 110 min, the flank face of Tool B reached the specified wear width, whereas the frictional coefficient and sliding energy of Tool B changes were still flat. 

### 3.3. Abrasive Wear of Tool A and Tool B 

As can be seen from Figure 10(IA,IB), the narrow area near the cutting edge of the rake face of Tool A was brighter than the other areas. This is because the color of the coating was dark, which indicates that the abrasive wear was serious and the coating material of the tool was almost polished. At the same time, there were a lot of clear plowing lines on the rake face, indicating that the friction between the tool and the chip was severe. As can be seen from Figure 10(IC,ID), the brightening narrow area near the cutting edge of Tool B was shorter and less wide than that of Tool A and the width of the narrow area was unevenly distributed. The plowing lines of the rake face were sparse and thus the abrasive wear was slight. 

Figure 10II shows the wear of the flank face of Tool A and Tool B. There was a severe wear zone in the near-field of the cutting edge on the flank face of Tool A and the plowing lines were also densely distributed. As can be seen from Figure 10(IIC,IID), there was wear in the same area of Tool B as compared with Tool A. However, the average wear width was smaller. There were fewer plowing lines in the lower part of the wear zone and there were shallower wear marks. The abrasive wear on the flank face was slight.

### 3.4. Comparison of Adhesive Wear and Oxidation Wear of Rake Face Tool A and Tool B

In the cutting process of AISI 304, when the chips passed through the rake face, the contact area between the tool and the chip was under high-temperature and high-pressure conditions. The adhesive force and the stick–slip effect caused stress and fractures on the coating surface, wearing the coating at last. This made the chip contact with the tool directly, and thus aggravated the wear.

The SEM and EDS analysis results of the worn rake face of Tool A are shown in Figure 11 and Figure 12, respectively. We noted from Figure 11A–C that the coating exfoliation in the near-field of cutting edge was serious. There was obvious adhesive on the rake face and there was a large piece of adhesive far away from the tool tip where chips outflowed and the residual chips stuck easily. Figure 11D–F shows the surface coating materials of the tool, which are Ti, Al, and N, respectively. As can be seen from the figure, the three elements were mainly distributed in the area with slight tool wear. Figure 11G shows Fe which was derived from the workpiece. During the cutting process, due to the severe friction between the chips and the tool, the Fe element of the chip was partly transferred to the tool and thus adhesive wear occurred. The O element is shown in Figure 11H. There was no O in either the cutting tool or the workpiece. Therefore, there was a chemical reaction between oxygen in the air and the element attached to the rake face. W was mainly distributed in the area near the cutting edge (Figure 11I), which indicated that the surface material of the tool in this area had been polished and the underlying material was exposed.

The main elements of point P1 in Figure 11B and point P2 in Figure 11C were mapped to Figure 12 by EDS analysis. It can be seen from Figure 12A that the content of W in this area was the highest. Such a result indicates that the surface material of the tool produced serious peeling, exposing the matrix material of the tool. The contents of O, Fe, and Cr were relatively high, which is consistent with the composition of AISI 304 in Table 3. In addition, during the cutting process, adhesive wear and oxidation wear occurred due to the existence of O in the air. As shown in Figure 12B, Fe at P2 was the highest, followed by O, Cr, and W. Such a result manifested in severe adhesive wear as well as oxidation wear in this area.

As a comparison, we also performed the SEM and EDS analysis on the rake face of Tool B (Figure 13 and Figure 14). By comparing Figure 11A,B and Figure 13A,B, it was found that the area near the cutting edge of Tool B was narrower and nonuniform. Furthermore, there was no large adhesive on the rake face and there was a smaller adhesive wear area compared with Tool A. We noted from Figure 13D–F that the distribution of Ti, Al, and N on the rake face of Tool B was denser and the coverage was wider compared with Tool A (Figure 11D–F), which indicated that the wear of the tool coating was less. Moreover, the Fe and O distribution of Tool A was more compact than Tool B with a higher area ratio (Figure 11 and Figure 13G,H), indicating that adhesive wear and oxidation wear of Tool A were more serious.

Compared with the distribution of element W (Figure 11I and Figure 13I), the distribution width of W in Tool B was narrower, sparse, and fluctuant. Such a result resulted in lower surface wear of the material at the bottom of Tool B, and thus the explosion of the bottom material was less than that of Tool A. Then, we mapped the main elements of point P1 in Figure 13B and point P2 in Figure 13C to Figure 14 with EDS analysis. As shown at point P1 of Tool B in Figure 14A, Ti and Al (coating materials of the tool) accounted for a large proportion. In contrast, the highest content of element at the corresponding point of Tool A was W (Figure 12A). Other elements, like O, Fe, and Cr, also occupied a large proportion. Therefore, we can conclude that this area was slightly worn and there was slight adhesive wear and oxidation wear for Tool B. Similarly, the lower content of Fe and Cr shown in Figure 14B also manifested in lower degree of adhesive wear in Tool B. The micro-groove design of Tool B changed the tool–chip contact state from full contact to incomplete contact, resulting in decreases of the stress, cutting forces, temperatures, and tool wear.

#### 3.5. Comparison of Adhesive Wear and Oxidation Wear of the Flank Faces of Tool A and Tool B

SEM and EDS analysis results of the flank face of Tool A are shown in Figure 15 and Figure 16. From Figure 15A–C, it can be seen that the surface material near the cutting edge of the flank face of Tool A was seriously peeled. Near the tip of the tool, the size of the adhesive was larger, which led to serious wear. Figure 15D–F shows the distribution area and density of the coating material of Ti, Al, and N, respectively. We found that the wear near the cutting edge of the flank face was serious. Moreover, we noted from Figure 15A,G,H that Fe and O were mainly distributed on whitening and uneven areas of the flank face, which was the mixed zone of adhesive wear and oxidation wear. In Figure 16I, W mainly distributed in the area near the cutting edge, indicating that the tool coating was almost polished to show the underlying material. We also chose two points (P1 and P2) on the flank face of Tool A with serious wear for analysis by EDS (Figure 16). Concentrations of Fe, Cr, and O were relatively high for both points, which is consistent with the intuitive result shown in Figure 15A that the adhesive wear and oxidation wear at this point were serious.

In contrast, we also show the results of the flank face of Tool B with SEM and EDS analysis in Figure 17 and Figure 18. It can be seen from Figure 17A–C that the surface material in the cutting-edge neighborhood of the flank face of Tool B was badly worn. However, compared with the corresponding position of Tool A, the wear width was narrower with weaker adhesive. By comparing distributions of Ti, Al, and N between Tool A and B, the coating material distribution area of Tool B was larger and the concentration was higher. Such a result indicates that the abrasive wear of Tool B was less. Furthermore, the sparse distribution of Fe and O on Tool B as compared to that of Tool A also resulted in a slight degree of adhesive wear and oxidation wear (Figure 17G,H).

We also analyzed two worn points on Tool B using EDS (Figure 18). Concentrations of Fe, O, and Cr at both P1 and P2 on Tool A (Figure 16) were higher than those on Tool B (Figure 18), indicating that the adhesive and oxidative wear of Tool A were more serious as compared with Tool B.

The friction of the tool–chip area was mainly composed of two parts. The first was the internal friction area near the tool tip which was affected by high temperature and high pressure. The material at the bottom of the chip had serious plastic deformations and was bonded to the material at the rake face of the tools. This area was also known as the adhesive friction area, and it had great friction resistance. The second was the tool–chip contact area, which was far away from the external friction area of the tool tip. The pressure and temperature in this area were lower and subject to Coulomb’s law of friction, known as the sliding friction area. The placement of the micro-groove of Tool B changed the contact state between the tool and the chip. As can be seen from Figure 19, due to the existence of the micro-groove, when the chip flowed into the micro-groove from the rake face, the chip flow suffered less resistance and the chip deformation was smaller. Chips did not come in contact with the bottom of the micro-groove while flowing through the micro-groove. When chips were about to flow out of the micro-groove, they were hindered by the micro-groove edge which was far away from the main cutting edge near the tip of the tool; thus, the chips curled out. At the same time, the micro-groove created a space for chips to flow freely.

We made 2-dimentional simulation of tool A and tool B with Deform V11.2. The parameter setting and material model are consistent with the tool temperature simulation. As can be seen in Figure 19, the tool–chip contact length of Tool B was reduced in comparison to Tool A. The average chipping thickness of Tool A was 0.36 mm, whereas the average chipping thickness of Tool B was 0.25 mm. The chipping thickness of Tool B decreased with the decrease of the energy consumed by chip deformation. Therefore, the overall cutting resistance, the cutting energy, the cutting temperature, and the tool wear of Tool B decreased. This decrease was compared with that of Tool A.

Due to the existence of micro-groove on the rake surface of tool B, the contact area between the tool and the chip is actually reduced, thus reducing the friction of the chip flowing through the rake surface as well as the cutting temperature of the tool. High temperature is the most important factor for adhesive wear and oxidation wear. Therefore, the adhesive wear and oxidation wear of the rake and flank surfaces of tool B are less than that of tool A. Kang et al., by laser processing different texture on the rake face of tool, found in the cutting process, the cutting force is reduced and the cutting stability is improved [33]. Thomas et al., using the indentation method to make texture on the rake surface, found that micro-texture reduces cutting force and cutting temperature and tool wear [34]. These are consistent with the structure of this paper.

## 4. Conclusions

To improve tool wear performance, a micro-groove was designed on the rake face of the tool. We compared the cutting performance of AISI 304 with Tool A and the micro-groove turning Tool B, both theoretically and experimentally. Results show that placement of the micro-groove made Tool B have better cutting performance. Specific conclusions are as follows.

Compared with Tool A, the placement of the micro-groove made the rake angle of Tool B increase whereas the friction angle, the cutting force, and the cutting temperature decreased.According to the surface morphology, the abrasive wear of the rake and flank faces of Tool B was less than that of Tool A. At the same time, the distributions of the main elements and the energy spectrum analysis of the local areas show that the adhesive wear and oxidation wear of Tool B were also less than those of Tool A.Due to the existence of the micro-groove on the rake face of Tool B, the tool–chip contact state changed and the length of the tool–chip friction zone was reduced. Meanwhile, it provided more stretching space for chip movement and reduced the energy consumed by chip deformation.

## Figures and Tables

**Figure 1 materials-13-01236-f001:**
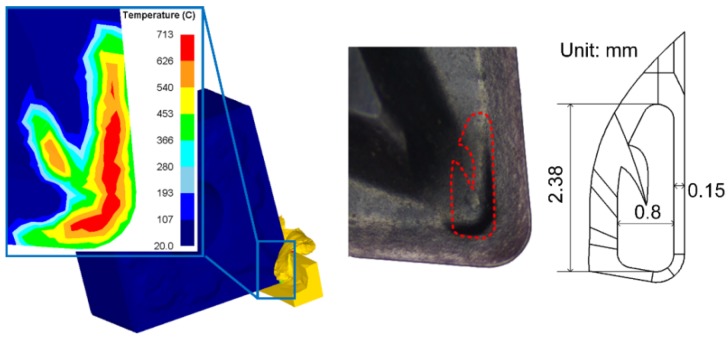
Shape of micro-groove tool and geometric dimensions.

**Figure 2 materials-13-01236-f002:**
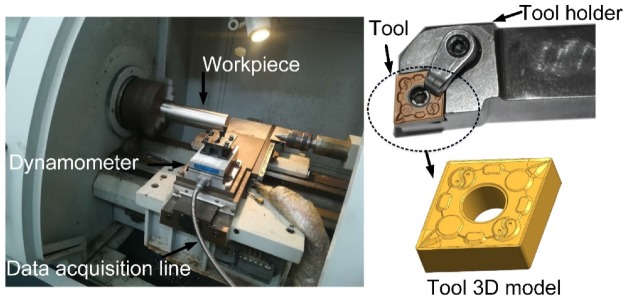
Turning test device and tool 3D model.

**Figure 3 materials-13-01236-f003:**
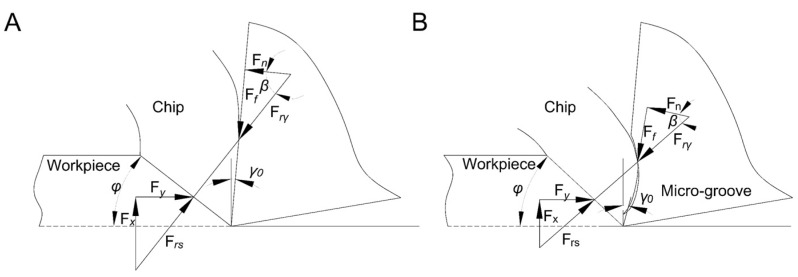
Tool force balance relationship in two-dimensional cutting. (**A**) Tool A. (**B**) Tool B.

**Figure 4 materials-13-01236-f004:**
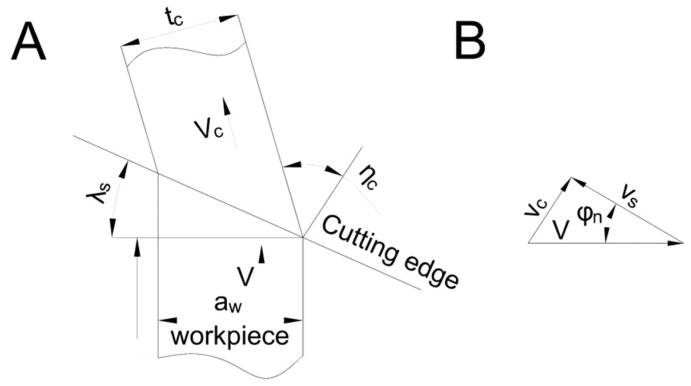
(**A**) Bevel cutting model. (**B**) Velocity vector triangle.

**Figure 5 materials-13-01236-f005:**
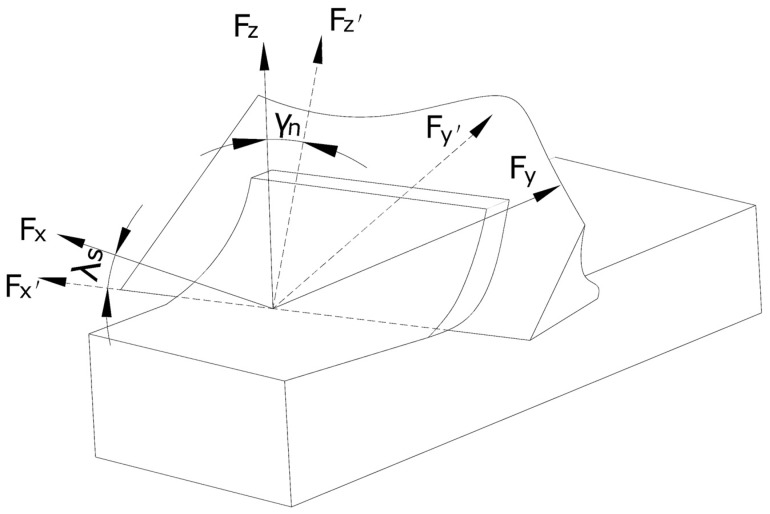
Conversion model of bevel cutting and right angle cutting.

**Figure 6 materials-13-01236-f006:**
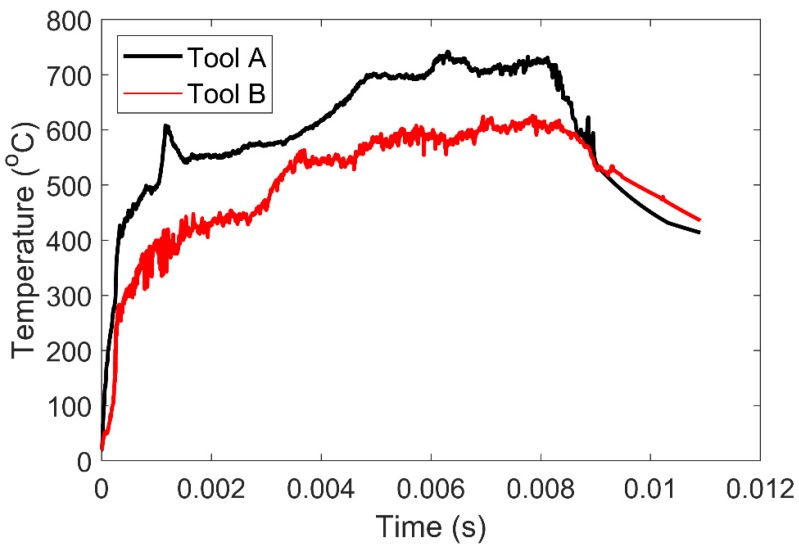
Cutting simulation temperature of Tool A and Tool B.

**Figure 7 materials-13-01236-f007:**
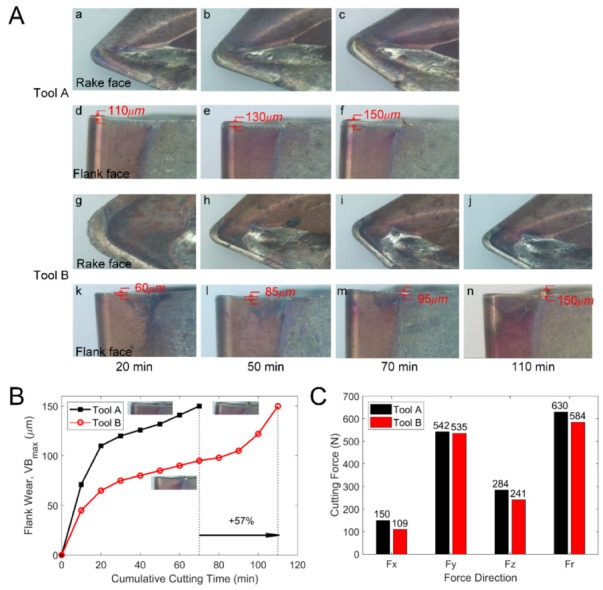
(**A**) The comparison of wear morphology of the rake and flank faces of the two tools. (**B**) The wear curve of the two cutting tools in processing AISI 304. (**C**) The comparison of the average cutting force and resultant cutting force of the two tools.

**Figure 8 materials-13-01236-f008:**
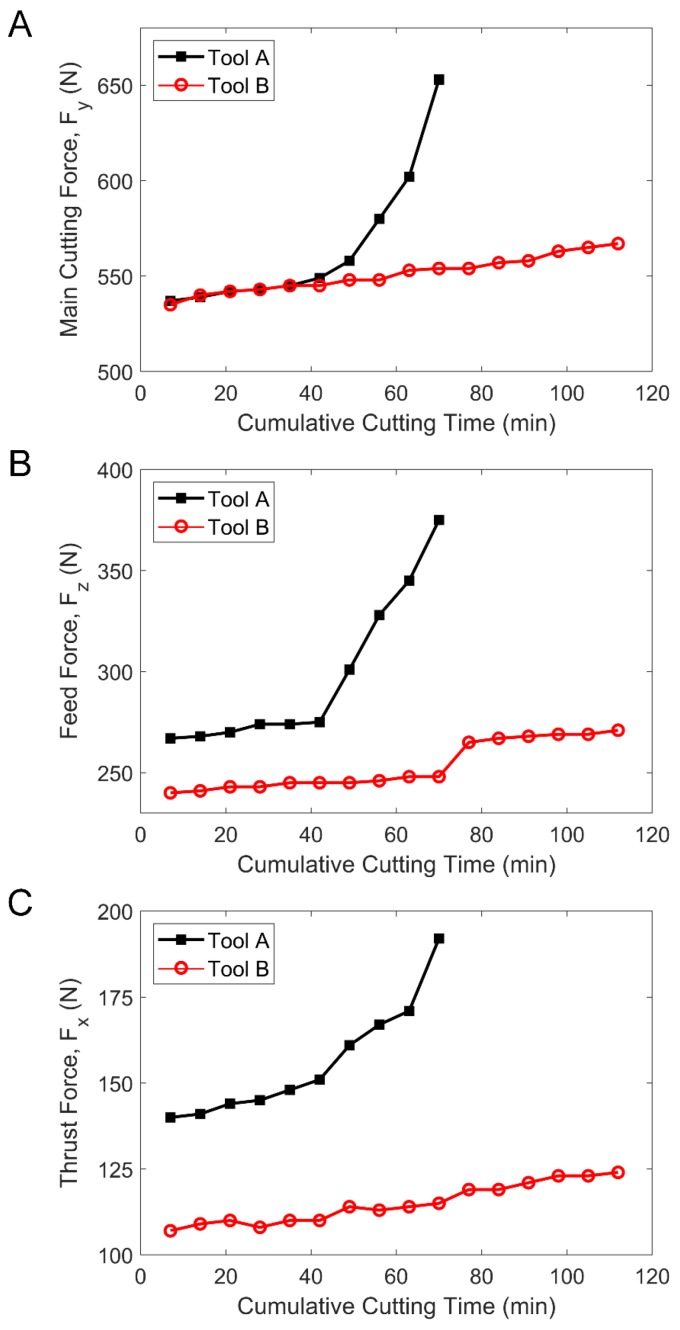
The comparison curves of the cutting forces of Tool A and Tool B over time. (**A**) The main cutting force. (**B**) Feed force. (**C**) Radial thrust force.

**Figure 9 materials-13-01236-f009:**
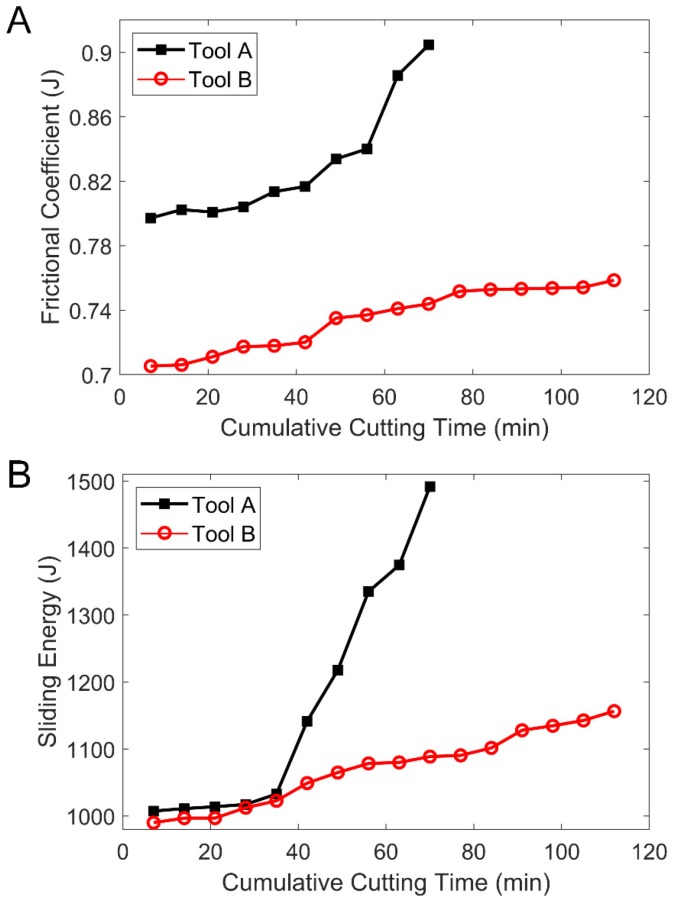
(**A**) Comparison curves of frictional coefficients of Tool A and Tool B over time. (**B**) The sliding energy curves of Tool A and Tool B over time.

**Figure 10 materials-13-01236-f010:**
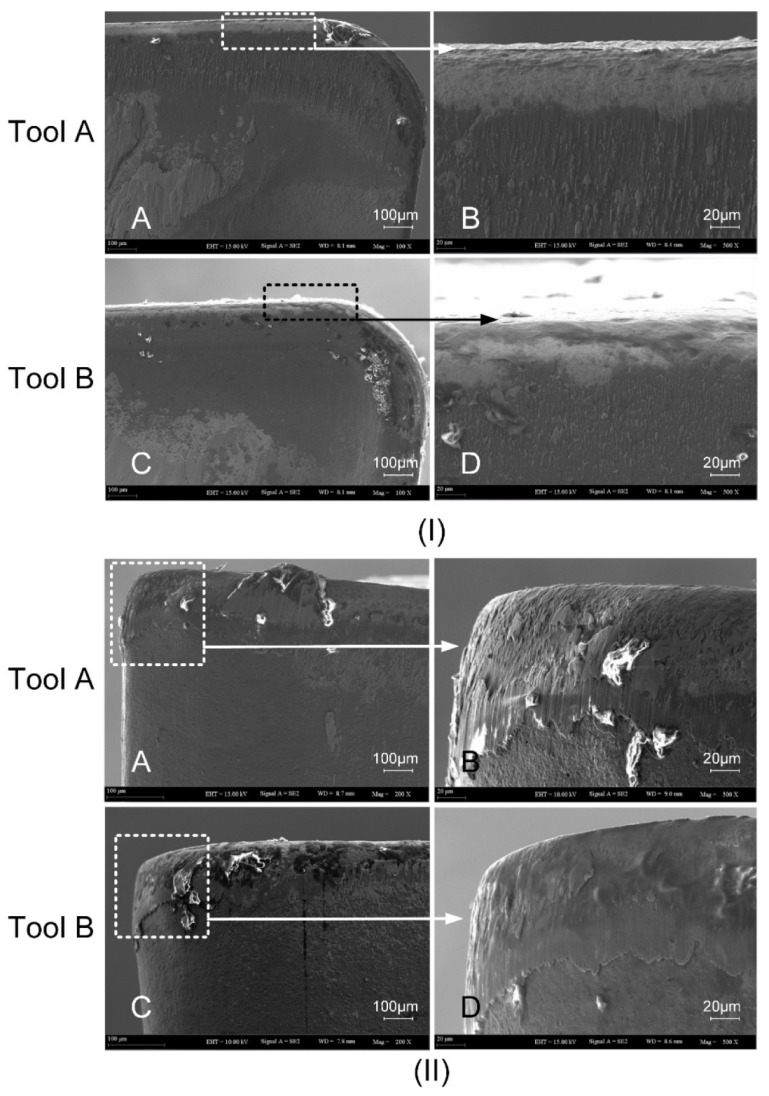
(**I**) SEM diagram of Tool A and Tool B wear on the rake face. (**II**) SEM diagram of Tool A and Tool B wear on the flank face.

**Figure 11 materials-13-01236-f011:**
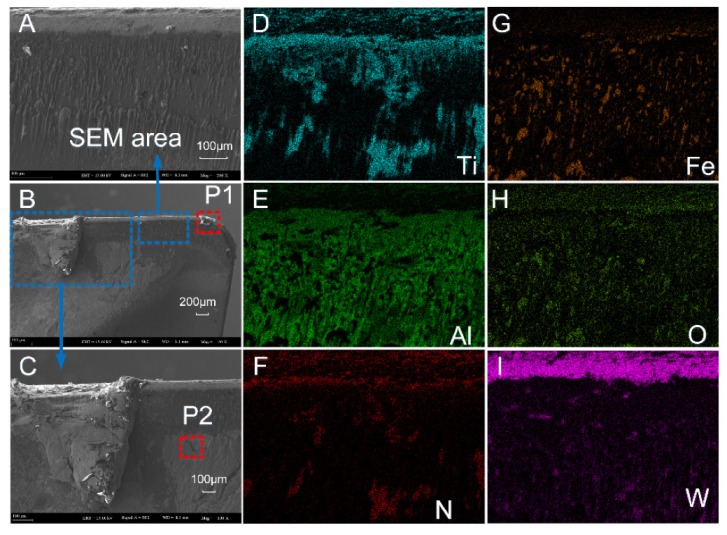
(**A**–**C**) The rake face wear of Tool A after 70 min of cutting. (**D**–**I**) Distribution of elements.

**Figure 12 materials-13-01236-f012:**
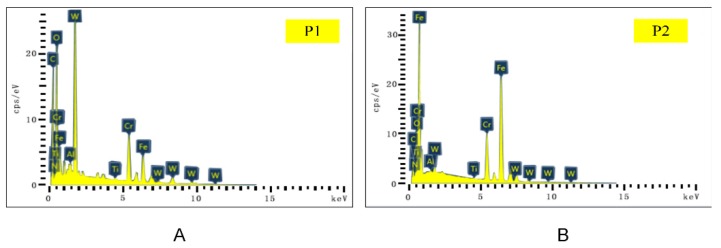
Energy dispersive X-ray spectroscopy (EDS) analysis results of corresponding points P1 (**A**) and P2 (**B**) in Figure 11.

**Figure 13 materials-13-01236-f013:**
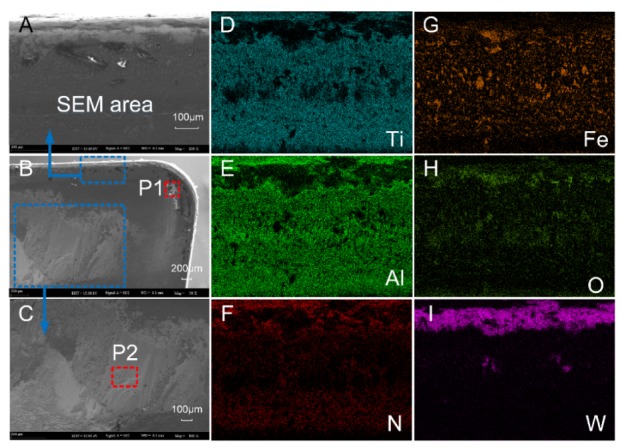
(**A**–**C**) The rake face wear of Tool B after 110 min of cutting. (**D**–**I**) Distribution of major elements.

**Figure 14 materials-13-01236-f014:**
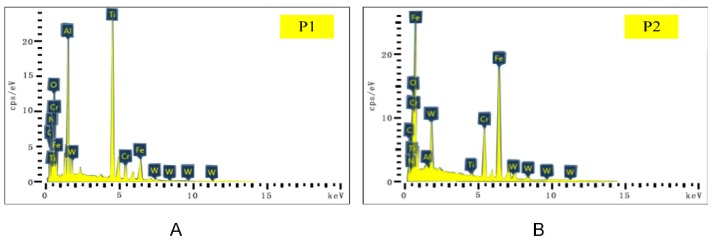
EDS analysis results of corresponding points P1 (**A**) and P2 (**B**) in Figure 13.

**Figure 15 materials-13-01236-f015:**
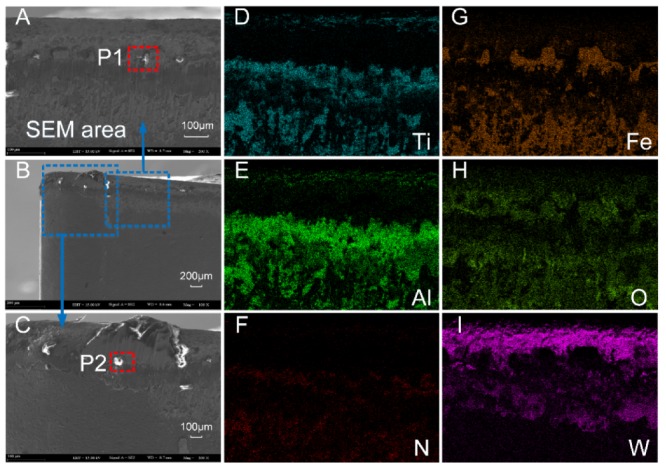
(**A**–**C**) The flank face wear of Tool A after 70 min of cutting. (**D**–**I**) Distribution of main elements.

**Figure 16 materials-13-01236-f016:**
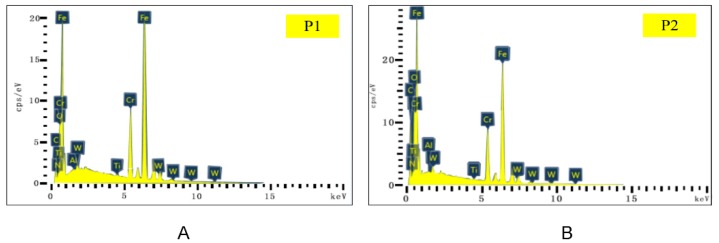
EDS analysis results of corresponding points P1 (**A**) and P2 (**B**) in Figure 15.

**Figure 17 materials-13-01236-f017:**
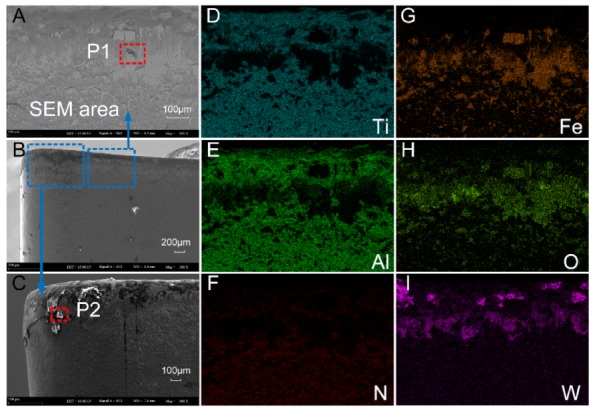
(**A**–**C**) The flank face wear of Tool B after 110 min of cutting. (**D**–**I**) Distribution of major elements.

**Figure 18 materials-13-01236-f018:**
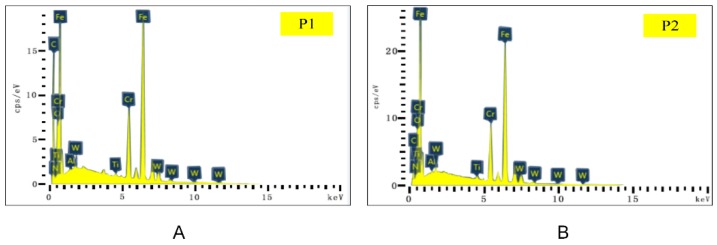
EDS analysis results of corresponding points P1 (**A**) and P2 (**B**) in Figure 17.

**Figure 19 materials-13-01236-f019:**
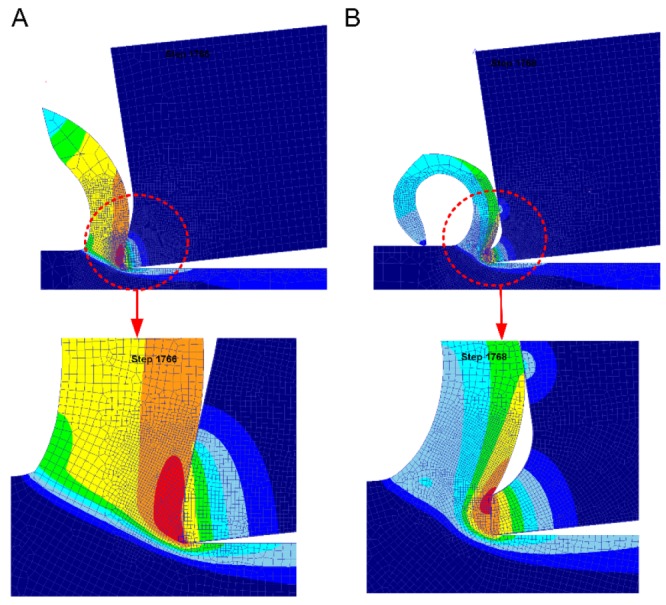
(**A**) Tool–chip contact diagram of Tool A. (**B**) Tool–chip contact diagram of Tool B.

**Table 1 materials-13-01236-t001:** Geometric angles of the tools.

Geometric Angle	Value (°)
Tool Angle *ε_r_*	80
Rake angle *γ_0_*	8
Clearance angle *α_0_*	7
Main cutting edge angle *K_r_*	95
End cutting edge angle *K_r_*’	−5
Inclination angle *λ_s_*	−5

**Table 2 materials-13-01236-t002:** Physical properties of tool and workpiece.

Target Object	Tool	AISI 304	Unit
Density *ρ*	14.6	7.93	g/cm^3^
Tensile strength	784.5	0.23	MPa
Poisson Ratio *μ*	0.23	0.247	
Hardness	89.5HRA	29HRC	HRA or HRC
Elasticity Modulus	634	206	GPa

**Table 3 materials-13-01236-t003:** Chemical composition of AISI 304 (wt %).

Si	Mn	P	S	Ni	Cr	C	Fe
0.75	1.64	0.045	0.03	8.56	18.87	0.08	70.025

**Table 4 materials-13-01236-t004:** Flank wear of Tool A.

Time (min)	20	50	70
Test 1: Flank wear, VB (μm)	112	133	151
Test 2: Flank wear, VB (μm)	108	132	148
Test 3: Flank wear, VB (μm)	110	132	150
Average (μm)	110.0	132.3	149.7
Standard deviation	1.63	0.58	1.53

**Table 5 materials-13-01236-t005:** Flank wear of Tool B.

Time (min)	20	50	70	110
Test 1: Flank wear, VB (μm)	65	83	96	148
Test 2: Flank wear, VB (μm)	64	86	93	154
Test 3: Flank wear, VB (μm)	65	85	95	150
Average (μm)	64.7	85.0	94.7	150.7
Standard deviation	0.58	1.53	1.53	3.06

**Table 6 materials-13-01236-t006:** The cutting force at the 40 min of the experiment for Tool A.

Time (min)	40			
Cutting Force (N)	F_x_	F_y_	F_z_	F_r_
Test 1	150	542	280	629.8
Test 2	148	540	289	630.1
Test 3	150	542	284	630.0
Average	149.3	542.3	284.3	630.0
Standard deviation	1.15	1.15	5.15	0.15

**Table 7 materials-13-01236-t007:** The cutting force at the 40 min of the experiment for Tool B.

Time (min)	40			
Cutting Force (N)	F_x_	F_y_	F_z_	F_r_
Test 1	107	533	250	586.2
Test 2	112	535	246	585.1
Test 3	109	535	248	584.0
Average	109.3	534.8	248.0	585.5
Standard deviation	2.52	1.15	2	1.1

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
