# Peer review of "Tribological Performance of Micro-Groove Tools of Improving Tool Wear Resistance in Turning AISI 304 Process"

_materials, 2020, doi:10.3390/ma13051236_

Round 1
Reviewer 1 Report
The problem statement isn't clear.
The industrial application of this work needs to be highlighted.
More recent references about machining difficult-to-cut materials should be added:
Effects of nano-cutting fluids on tool performance and chip morphology during machining Inconel 718
Application of acoustic emissions in machining processes: analysis and critical review
Sustainability assessment associated with surface roughness and power consumption characteristics in nanofluid MQL-assisted turning of AISI 1045 steel
Figure 17 for Tool-chip contact diagram of Tools A & B isn't clear
A tribological mechanism should be given to physically justify all tool wear results
Reviewer 2 Report
This very interesting paper deals with the effect of using new type of tool with micro-groove when turning AISI 304 material. The wear of the cutting edge and the cutting forces were measured and discussed. This manuscript has good quality to be accepted but some details miss to understand exactly what Authors would like to show. I suggest to accept this manuscript after revisions regarding to the following comments:
In the Materials and methods:
- In the article there is no precise information about the cutting tool. No information about the material of the cutting insert. Has the tool used in the tests had a chip breaker? What was the radius of the cutting insert? What was the depth of the groove for tool B and what was the shape of the chip breaker before and after the micro-groove was made? Ware this data analyzed in the simulation? What material model was adopted in the simulation proprietary program (Johnson and Cook?).
- Was any research plan adopted and statistical analysis of results carried out?
In the Results and discussion:
- Fig. 2 shows the temperature dependence for a very short simulation time (i.e. max value 0.01 s). What was the temperature graph for the time during which the tool worked in a stable manner?
- Fig. 3a should be better quality.
- What was the dispersion of the results of wear (Fig 3 b,c) and cutting forces measurements?
- on Fig. 17 - Is it Thirdwave AdvantEdge or is it Deform software? If the Thirdwave program was used, it is necessary to supplement the information about the reason for its use, simulation parameters, material model, etc. No information presented in the description of the results from the line 324.
- There is no reference to the publication of the results achieved by other authors. These chapter should be combined with other published articles.
- In manuscript there are several wrong references to literature (i.e. information “Error! Reference source not found”). This is probably because a text editing program error.
